CUTTING EDGE

# Building bridges between cellular and molecular structural biology

**Abstract** The integration of cellular and molecular structural data is key to understanding the function of macromolecular assemblies and complexes in their in vivo context. Here we report on the outcomes of a workshop that discussed how to integrate structural data from a range of public archives. The workshop identified two main priorities: the development of tools and file formats to support segmentation (that is, the decomposition of a three-dimensional volume into regions that can be associated with defined objects), and the development of tools to support the annotation of biological structures.

ARDAN PATWARDHAN[*], ROBERT BRANDT, SARAH J BUTCHER,
LUCY COLLINSON, DAVID GAULT, KAY GRÜNEWALD, COREY HECKSEL[†],
JUHA T HUISKONEN, ANDRII IUDIN, MARTIN L JONES, PAUL K KORIR,
ABRAHAM J KOSTER, INGVAR LAGERSTEDT[‡], CATHERINE L LAWSON,
DAVID MASTRONARDE, MATTHEW MCCORMICK, HELEN PARKINSON,
PETER B ROSENTHAL, STEPHAN SAALFELD, HELEN R SAIBIL,
SIRARAT SARNTIVIJAI, IRENE SOLANES VALERO[§], SRIRAM SUBRAMANIAM,
JASON R SWEDLOW, ILINCA TUDOSE, MARTYN WINN AND
GERARD J KLEYWEGT[*]

**\*For correspondence:** ardan@ebi.ac.uk (AP); gerard@ebi.ac.uk (GJK)

**Present address:** [†]Electron Bio-Imaging Centre, Diamond Light Source Ltd, Didcot, United Kingdom; [‡]Computational Chemistry and Cheminformatics, Lilly UK, Windlesham, United Kingdom; [§]University of Vic - Central University of Catalonia, Barcelona, Spain

## Introduction

To obtain an integrated view of how molecular machinery operates inside cells, biologists are increasingly combining structural data at different length scales, obtained using a range of techniques such as electron tomography, electron microscopy, NMR spectroscopy and X-ray crystallography. Structural data is held in public archives such as the Electron Microscopy Data Bank (EMDB; emdb-empiar.org; *Tagari et al., 2002*), the Electron Microscopy Public Image Archive (EMPIAR; empiar.org; *Iudin et al., 2016*), and the Protein Data Bank (PDB; wwpdb.org; *Bernstein et al., 1977*)

Integration between PDB and EMDB data is based on atomic models in the PDB that have been fitted to or built into EMDB volume maps. For purified biological molecules or larger defined complexes this approach is done routinely. Sequence information from the models

can be used to link to other bioinformatics resources such as the Universal Protein Resource (UniProt; uniprot.org; *UniProt Consortium, 2013*). However, atomic models are not always available for a variety of reasons, such as when molecular averaging fails to obtain high-resolution features or the inherently lower resolution associated with molecules being imaged in more complex or even cellular environments. In such cases, the identification of features often relies on prior knowledge or correlation of structural data obtained at different scales.

Once features have been identified, segmentation (defined here as the decomposition of the 3D volume into regions that can be associated with defined objects) can be employed to facilitate and visualise the interpretation of the map. For example, in a recent study the segmentation of electron and soft X-ray tomography reconstructions was used to study leakage and

**Figure 1.** Segmentation of *Plasmodium falciparum*–infected erythrocytes. Soft X-ray tomography shows loss of mechanical integrity of the red cell membrane in the final stages of egress. Panels A-C depict schizonts treated with a selective malarial cGMP-dependent protein kinase G inhibitor (C2), and panels D-F depict schizonts treated with a broad-spectrum cysteine protease inhibitor, E64, which allows parasitophorous vacuole membrane (PVM) rupture but prevents erythrocyte membrane rupture, resulting in merozoites trapped in the blood cell. (**A**) Slice from tomogram of C2-arrested schizont. (**B**) Outlines of erythrocyte membrane (red), PVM (yellow), and parasites (cyan) in the tomogram slice in *A*. (**C**) 3D rendering of the schizont. The vacuole (yellow) is densely packed with merozoites (cyan) that have been collectively rather than individually rendered, for clarity. The overall height of the cell is ~5 µm. (**D**) Tomogram slice from an E64-arrested schizont, shown with outlining of membranes in **E**. Remnants of the PVM are visible. (**F**) 3D rendering of the schizont. Figure and legend adapted with permission from *Hale et al. (2017)*. Scale bar 1 µm.

breakage of the membranes in erythrocytes infected by *Plasmodium falciparum*, and documented the dramatic changes in the morphology of cells during egress (*Hale et al., 2017*). The soft X-ray tomograms provided overviews of the membrane compartments in intact, vitrified cells (*Figure 1*). It should be noted that the word 'segmentation' may have different interpretations: for example, in whole animal, pre-clinical and medical imaging, segmentation includes a concept of a model that is used for fitting of the features. In this manuscript we limit the definition to the separation of density into distinct sub-domains.

In tomography, where multiple copies of nearly identical objects are found, 3D sub-tomogram averaging and 3D classification may be employed to obtain higher resolution reconstructions. This process often involves combining information from multiple tomograms. Since the higher resolution afforded by sub-tomogram averaging provides more structural detail, displaying sub-tomogram averages at the original tomogram positions and orientations may reveal important information about the organization and distribution of the object within a cellular and functional context. If properly annotated such data can be further mined with other questions in mind by other researchers. For example, researchers recently created composite maps of Lassa virus particles by inserting the sub-tomogram average structure of the Lassa virus glycoprotein spike back into the original tomographic reconstructions, revealing the organisation and copy number of the spikes on the virus surface (*Figure 2*; *Li et al., 2016*). Another example revealed the lateral clustering of viral membrane proteins mediating membrane fusion (*Maurer et al., 2013*).

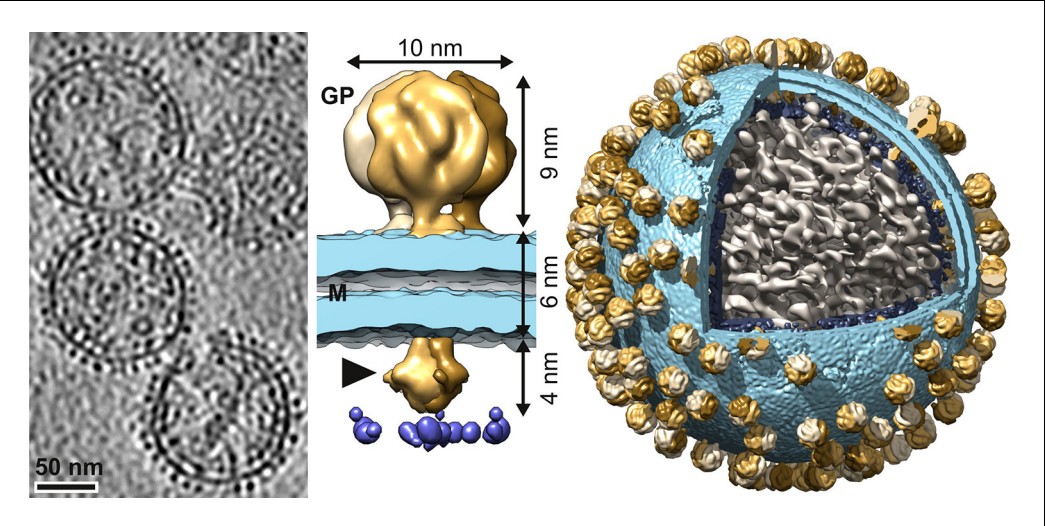

**Figure 2.** Arrangement of Lassa virus glycoprotein spikes on the virion surface. Left to right: A slice from a tomographic volume of Lassa viruses, a sub-tomogram average of the glycoprotein spike, and the sub-tomogram average inserted back onto a virus reconstruction. Images adapted from *Li et al., 2016* (under a CC BY 4.0 license).

The archiving of segmentation data in EMDB entries was identified as an area requiring urgent attention in previous workshops on "Data-Management Challenges in 3D Electron Microscopy" in 2011 (*Patwardhan et al., 2012*) and "A 3D Cellular Context for the Macromolecular World" in 2012 (*Patwardhan et al., 2014*), as was the improved biological annotation of structural data to make it more accessible to the wider biological audience and to enable integration with structural and other bioinformatics resources. Crucially for data integration we need "structured biological annotation" which is here defined as the association of data with identifiers (e.g., accession codes from UniProt) and ontologies taken from well established bioinformatics resources. (Ontologies are formal collections of statements defining concepts, relationships and constraints; for example, the mitochondrial large and small ribosomal subunits are parts of the mitochondrial ribosome which, in turn, is a part of the mitochondrion). To our knowledge, none of the segmentation formats widely used in electron microscopy and related fields currently support structured biological annotation. Furthermore, spatial transformations relating sub-tomograms to their parent tomograms are not currently captured in EMDB. Moreover, wider usage of both segmentation and transformation data by non-expert users is hindered by a plurality of formats.

To discuss and address the challenges of representing and capturing segmentations and transformation data, the Protein Data Bank in Europe (PDBe) organised an expert workshop on "3D Segmentations and Transformations - Building Bridges between Cellular and Molecular Structural Biology" in December 2015. The objectives were:

- To identify data models and formats for representing segmentation and transformation data that could provide support for structured biological annotation, thus facilitating their use by EMDB and enabling data-exchange between different software packages
- To gain a better understanding of the challenges involved in the annotation of electron microscopy data and develop requirements in terms of tools and strategies to facilitate annotation.

Here we report and discuss the main outcomes of the workshop, which was attended by a range of participants including software developers, users of segmentation software, ontology experts, and experts in structure and data archiving.

## Data models and file formats for segmentations and transformations

Prior to the workshop, PDBe developed a draft data model to support segmentations and their annotations in EMDB that could accommodate

segmentation descriptions from a range of existing formats and software packages as well as structured biological annotation. It supported the key features of major segmentation packages such as Amira (www.fei.com/software/amira), IMOD (*Kremer et al., 1996*) and Chimera (*Pettersen et al., 2004*), and provided scope for extension and flexibility as the field developed. However, the draft data model did not cover minor features (e.g., surface rendering parameters), especially those that are only relevant in the context of a particular software package. The data model was implemented in an XML schema with the following features:

a. *Support for hierarchical segmentation description.* This is important for representing segmentations from (semi-)automatic approaches that naturally result in a hierarchal segmentation, such as Segger (which iteratively groups the results of the initial watershed segmentation into a hierarchy; *Pintilie et al., 2010*).

b. *Different representations of segmentations.* Contours and simple geometric primitives such as spheres and lines are often used to delineate regions of interest (ROIs) when segmentation is performed manually. In automatic segmentation the segments are typically represented as surface meshes and/or 3D volume masks. In the latter case, run-length encoding and limited bit-depth are commonly used techniques to minimise memory requirements. It could be argued that it would be useful to have only one canonical representation and convert all the individual representations to it. However, representing geometric primitives such as spheres as surface meshes could lead to substantial increases in storage size and decreases in accuracy of the descriptions.

c. *Support for externally defined (i.e., as separate files) 3D volume masks.* It may be useful to allow separation between the metadata (annotations) and the actual segmentations (e.g., to lessen the burden on tools and web-services that only require the metadata). The data model accommodates links to external files (and locations within these files) for representing segments.

d. *Segment colours.* In some application areas, colour is used to identify objects of the same kind, so it is important that such information is not lost.

The draft data model was intended primarily for internal use in EMDB. However, the meeting participants strongly favoured a broader scope so that the format could serve the entire biological segmentation field. This would also make it easier to support the development of translators between different formats and possibly contribute to a reduction of the number of formats (or at least prevent further proliferation of formats). Representatives for several major software packages used for segmentation including IMOD (D. M.), Amira (R.B.) and Chimera (Tom Goddard, personal communication) have expressed a commitment to providing read/write capabilities for the developed format if standard libraries are made available.

The draft data model included support for various colour models including RGB, HSV and colour names. Participants argued that it would be sufficient to support only the most commonly used one, namely the RGB model, as the other models can be converted to it.

Participants also noted that it might be useful to allow quantification of the estimated certainty of a biological annotation, for example a score for the agreement between a sub-tomogram average and a corresponding region from an originating tomogram. There may also be alternative biological annotations in various combinations (logical OR, XOR, AND, etc.). The quantification of alternative annotations could become very complex to represent and use, and the participants agreed to initially limit the scope to a single annotation per segment and to let the need for more complex representations be driven by actual use cases.

Concerning the transformations between sub-tomogram averages and tomograms, the participants agreed that this information should be incorporated into the segmentation data model; it simply requires adding support for multiple transformations of the same 3D volume representation. It was agreed that the convention to define affine transformations should be well-defined in terms of the transformation, the order in which they are applied, the direction of the transformation, and the orientations and origins of coordinate systems.

With respect to correlative multi-modal imaging it was recognized that there would eventually be a need to go beyond affine transformations, for example to represent distortions and deformations of slice data, but the participants did not come to a conclusion about a coherent extensible format. Often, a segment consists of multiple spatially transformed copies of the same primitive. This is also relevant for sub-tomogram averages as the same volume is

to be spatially transformed into multiple locations within a tomogram. To accommodate these situations, every segment can be associated with a list of transformations. This representation will also be useful in the context of template matching for describing the transformations between the template and the 3D volume.

The draft data model was developed in XSD (XML Schema Definition). The definition of data models is greatly facilitated by tools that enable GUI-based development of schemas such as Oxygen and XMLSpy. Code generators such as generateDS create object-model wrappers from schemas that enable reading, writing and manipulation of XML files, thus allowing for rapid prototyping. Various XML validators also allow the correctness of a file relative to a schema to be tested. However, concerns were raised about the verbosity of the XML format and the efficiency with which it can be used. Participants proposed that while XML may be the natural format for a schema defined in XSD, it would be useful to consider other more compact and efficient formats such as JSON and HDF5 (a binary format that allows for efficient representation of hierarchal metadata and data in a single container). Both JSON and HDF5 are now widely supported with libraries in most major programming languages, including Python and C/C++, to facilitate reading and writing. To this end, utilities to convert between the XML, JSON and HDF5 representations of the segmentation data model are currently in development at PDBe.

Future format development will be an iterative process involving extensive consultation with relevant stakeholders to obtain consensus in and support from the community of developers, yielding a format that they will support. A "Segmentation and transformation file format working group" has been established by a subset of the workshop participants, and other developers working on segmentation who are interested in joining the group are asked to contact AP.

PDBe has already modified the data model based on the feedback from the meeting, and this will continue in several rounds of consultation with the working group. The schema is versioned to keep track of changes. To facilitate adoption of the format, dubbed EMDB-SFF (SFF=Segmentation File Format), PDBe is developing translators to/from other commonly used formats. The code for these translators is provided as free open source and distributed via the CCP-EM SVN repository. Comments on the schema should be sent to AP.

## Structured biological annotation

As previously explained, structured biological annotation is the association of data with identifiers and ontologies taken from well-established bioinformatics resources. The use of structured biological annotation is not common practice in the electron microscopy or structural biology communities. Therefore, ontology experts were invited to the workshop to explain why these are useful and what resources and tools are available for assigning annotations. Use-cases such as mouse imaging data helped to explain the principles and practice of structured biological annotation. By the end of the meeting there was a clearer appreciation of the importance of structured biological annotation for searching and linking imaging data across different scales, between different imaging and structural databases and with other bioinformatics resources.

Structured annotation would enable the seamless integration of structural, imaging and bioinformatics data from different resources, thus making it possible to provide problem-centric views of biology that incorporate structural and imaging data and are easily accessible by the broader biological community (and in contrast to the highly specialised structure-centric resources that are available today and mainly serve domain-specific communities). However, there were concerns that many in the electron microscopy community would find navigating the landscape of ontologies challenging and that this approach would only gain traction in the community if tools were developed to simplify the biological annotation process.

It was also discussed whether annotation should be performed by the depositor or by EMDB curators. While curators could be trained to a high level of expertise in the use of ontologies, they would not necessarily have enough knowledge about the sample and the specifics of the biological system underlying the study. It was concluded that depositors should perform the annotation, with curators overseeing and checking annotations.

## Tools for structured biological annotation

Structured biological annotation for electron microscopy will rely on a range of established ontologies such as Gene Ontology (GO; *Gene Ontology Consortium, 2008*),

Experimental Factor Ontology (EFO; *Malone et al., 2010*), Protein Ontology (PRO; *Natale et al., 2014*), Cellular Microscopy Phenotype Ontology (CMPO; *Jupp et al., 2016*), NCBI organismal classification (NCBITaxon), integrated cross-species for anatomical structures (UBERON; *Mungall et al., 2012*), imaging modality and sample preparation from Fbbi (*Orloff et al., 2013*), Foundational Model of Anatomy (FMA) and Cell Ontology (CL; *Diehl et al., 2016*). It may also include identifiers from resources such as UniProt and the Complex Portal, which in turn contain cross-reference information to other useful standardised vocabularies and common terminology identifiers, such as the OMIM and KEGG (*Kanehisa et al., 2016*) pathways. This cross-reference information is useful when linking data coded with these terminologies to the ontologies.

Several of these resources provide application programming interfaces (APIs) that can be used to access the information programmatically and provide search functionality. The Samples, Phenotypes and Ontologies Team (SPOT) at EMBL-EBI has developed tools such as Zooma and the Ontology Lookup Service (OLS; *Jupp et al., 2015*), which aggregate information from a wide range of ontologies and provide APIs to access these tools. These APIs can be used when building tools for segmentation annotation to provide simplified views and search facilities for ontological terms.

At the workshop, PDBe presented mock-ups of a web-based segmentation annotation tool (SAT; *Figure 3*). This tool would allow a user to add structured biological annotation to segmentations obtained from a variety of different software packages and then output an annotated segmentation file in EMDB-SFF that could be deposited to EMDB or EMPIAR. Annotation could either be done during deposition, in which case the biological annotation from the segmentation file could be harvested by the deposition system to facilitate the deposition process, or it could be done post deposition. The workflow would consist of: (i) the user uploading segmentation files (there could be several if the segments have been saved as separate files) and the corresponding map (unless it is already released in EMDB or EMPIAR); (ii) conversion to an EMDB-SFF file; (iii) use of a GUI-based interface to view the segmentations overlaid on the map and to select segmentations and add annotation; (iv) output of a fully annotated EMDB-SFF file that could be uploaded to EMDB (*Figure 4*).

Two different options were presented for how annotation could take place (*Figure 3*). Many macromolecular systems for which data are deposited in EMDB fall into broad categories such as ribosomes, proteasomes, chaperonins and so on: for each of these categories, and with the added information about taxonomy, lists of likely components could be generated to facilitate annotation (*Figure 3A*). Similarly for cellular level annotation, lists of cellular components could be used. As it would not be possible to cover every potential scenario with pre-defined lists, the other option is to provide a search facility that offers potentially applicable terms from available ontologies (*Figure 3B*).

The workshop participants expressed strong support for the development of the SAT and the functionality depicted in the mock-ups but raised concerns about a number of issues: the upload of data to a web server – some users may find it challenging to upload large maps and segmentations; the need to annotate segmentations twice – users would typically add free text annotations in the software used for the segmentation and would the need to re-annotate in the SAT; finding the 'right' metadata terms (particularly in cases where a search yields more than one term, and it is not clear which is the most relevant term); annotating a hierarchical segmentation. (The SAT mock-up accommodates annotation on only one level of hierarchy: this might be sufficient in many cases, but it could become problematic as more automated segmentation techniques are developed and their usage expands.)

A desktop version of the SAT would help users concerned about the upload of large amounts of data to a web-server. Another option would be to integrate the functionality for structured biological annotation into existing packages such as IMOD, Chimera and Amira; this would also avoid the problem of users having to annotate the segmentations twice. This alternative would require the development of libraries and widgets that facilitate the use of ontologies and the EMDB-SFF by third parties. For example, the program for segmentation in IMOD already has a 'Name Wizard' plugin that helps the user to choose standardized object names from a CSV file: however, additional development would be need to provide access to on-line ontologies.

Participants agreed that PDBe should start by developing the web-based SAT because it could reuse a number of components that are already being used in other electron microscopy-related

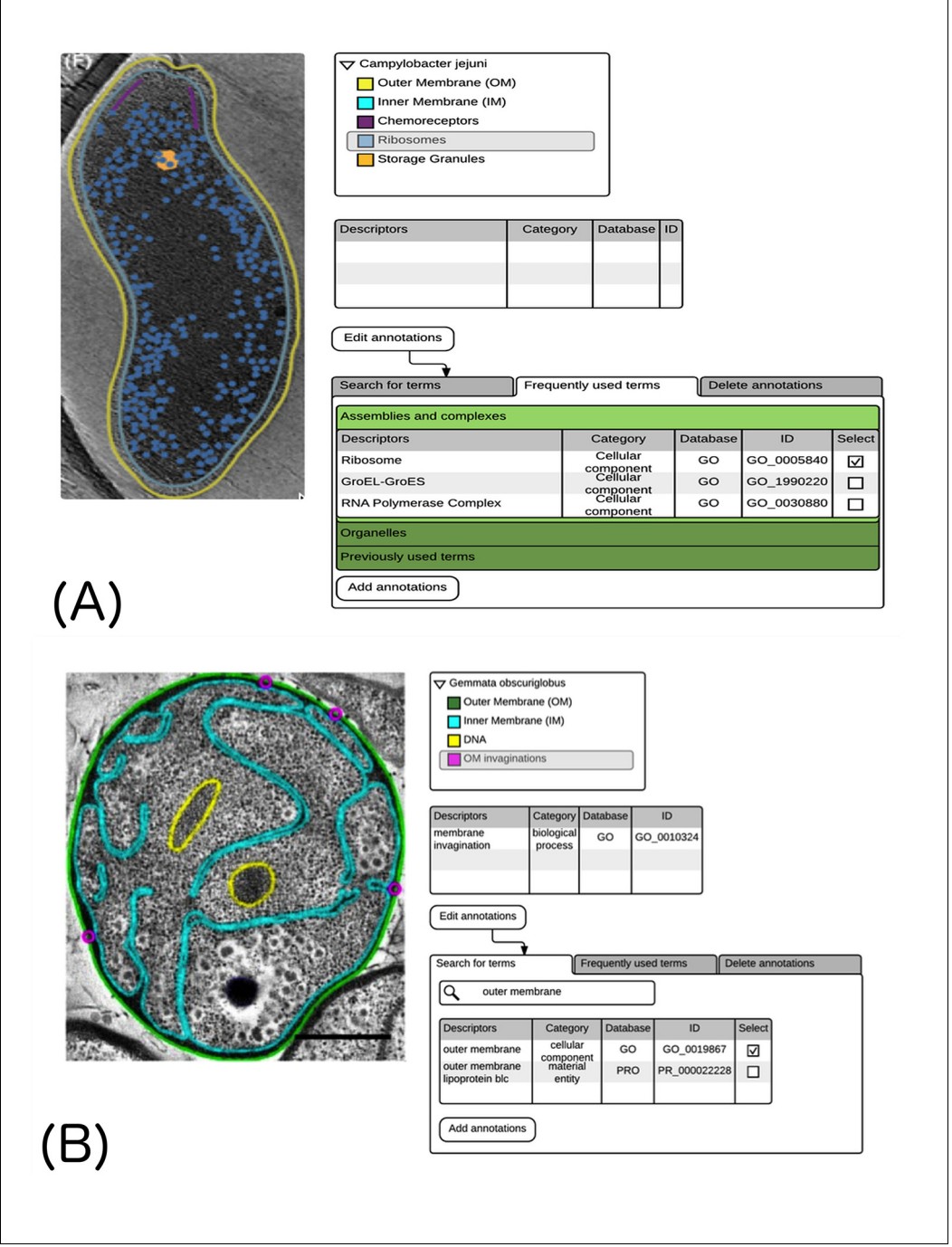

**Figure 3.** Mock-up of a possible Segmentation-Annotation Tool (SAT). Image slices are shown with the segmentations overlaid. (**A**) The top right panel presents a tree that enables the user to select the segment to be annotated, and existing annotations are shown in the middle right panel. The bottom right panel provides pre-defined lists of annotation terms for frequently studied assemblies and complexes. The image in the left panel is adapted from *Müller et al. (2014)* (under a CC BY 3.0 license). (**B**) The top right and middle right panels are similar to those in A. The bottom right panel provides a search option to find relevant terms. The image in the left panel is adapted from *Santarella-Mellwig et al., 2013* (under a CC BY 4.0 license).

web services (such as the Volume slicer; *Sala-vert-Torres et al., 2016*), followed by the desktop version. Once the SAT reaches a certain level of maturity PDBe could work with third-party developers to integrate the annotation functionality into their packages.

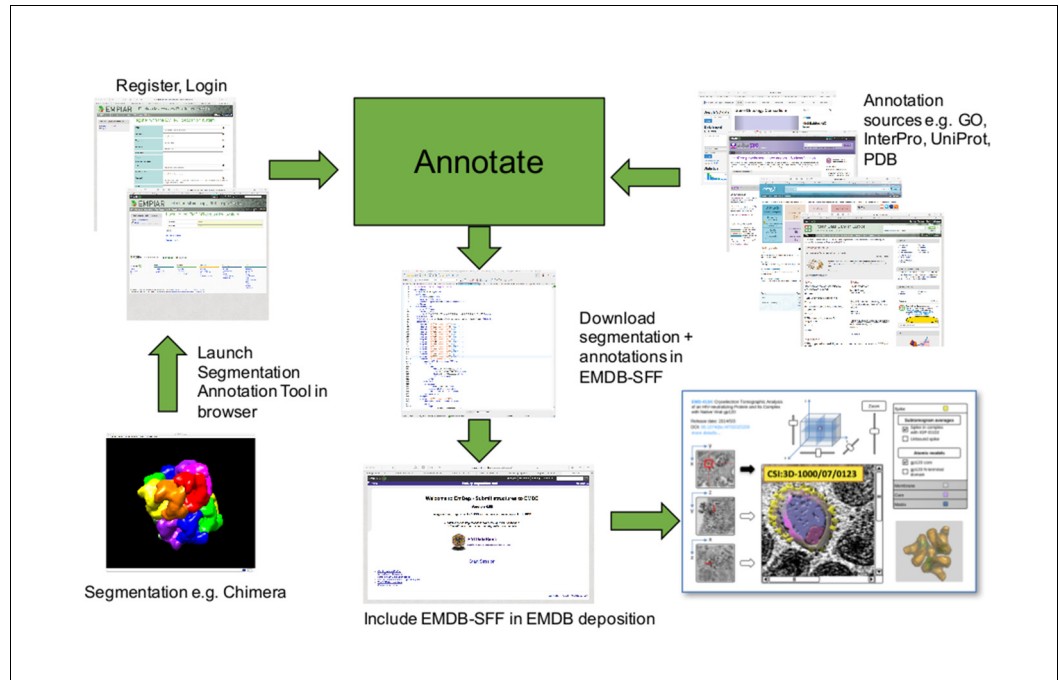

**Figure 4.** Segmentation-annotation workflow. A user launches the Segmentation-Annotation Tool and uploads segmentations obtained with third-party software. After the segmentation has been annotated with biologically meaningful terms, a segmentation file is written in EMDB-SFF format; this file can be uploaded to the Electron Microscopy Data Bank when the structure is deposited. Once released, the EMDB-SFF file can be used for the integration of structural data between different imaging scales and across resources. The Volume browser mock-up (bottom right) contains images adapted from *Bennett et al. (2007)* and *Bennett et al. (2009)* (under a CC0 1.0 license). The 3D rendering was generated from EMDB entry EMD-5020 and PDB entry 3dno (*Liu et al., 2008*).

By far the greatest challenge is developing the functionality to find the appropriate biological metadata (*Malone et al., 2016*) and tools such as Zooma and OLS will be useful for this purpose. A "Segmentation annotation working group" has been established by a subset of the workshop participants to provide data sets and use cases to aid the design of the SAT and to help with its testing. Members of the electron microscopy community and related communities who are interested in joining the group are asked to contact AP.

## Discussion

The EMDB-SFF data model has undergone a round of updates based on the feedback from the meeting. The development of the file format and the segmentation annotation tools will be iterative, with user-testing and feedback from the working groups being integral parts of the process. The file format and tools are expected to be ready by late 2017, although they might not offer all the features discussed above.

Wide acceptance and support of the EMDB-SFF format by software developers working on segmentations and transformations will be crucial. Providing well-documented open-source tools for working with the format will help in this regard, and the Collaborative Computational Project for Electron cryo-Microscopy (CCP-EM) has committed to distributing these tools, and including them in training events for users and developers. However, the scope of the format is not limited to the cryo-EM field. For example, segmentation is an essential element of the workflow for interpreting data in 3D scanning electron microscopy (3D-SEM; *Patwardhan et al., 2014*). It will also be possible to provide support for segmentations for other imaging modalities (and also for imaging on other length scales), although the range of biological ontologies and vocabularies will need to be expanded. It should also be possible to support techniques that combine imaging modalities (such as correlative light and electron microscopy), but this will involve extra work on the transformation model.

It was clear from the discussions regarding the annotation of segmentations that there are significant language barriers between the fields. Overcoming these barriers is a prerequisite for progress, as is the development of new tools that will facilitate annotation.

This workshop was an important milestone in that it defined concrete actionable outcomes to address the challenges involved in the integration of cellular and molecular structural data in the public archives. This integration will provide researchers with "problem-centric views" of data from many different sources, and will also help the wider biological and medical communities by making make structural data more accessible.

**Ardan Patwardhan** is in Cellular Structure and 3D Bioimaging, European Molecular Biology Laboratory, European Bioinformatics Institute, Wellcome Genome Campus, Hinxton, United Kingdom

http://orcid.org/0000-0001-7663-9028

**Robert Brandt** is in the Visualization Sciences Group, FEI, Mérignac, France

**Sarah J Butcher** is in the Institute of Biotechnology and the Department of Biosciences, University of Helsinki, Helsinki, Finland

**Lucy Collinson** is in the Electron Microscopy Science Technology Platform, Francis Crick Institute, London, United Kingdom

**David Gault** is in the Centre for Gene Regulation and Expression, University of Dundee, Dundee, United Kingdom

**Kay Grünewald** is in the Division of Structural Biology, Wellcome Trust Centre for Human Genetics, University of Oxford, Oxford, United Kingdom

**Corey Hecksel** is in the National Center for Macromolecular Imaging, Verna and Marrs McLean Department of Biochemistry and Molecular Biology, Baylor College of Medicine, Houston, United States

**Juha T Huiskonen** is in the Division of Structural Biology, Wellcome Trust Centre for Human Genetics, University of Oxford, Oxford, United Kingdom

**Andrii Iudin** is in Cellular Structure and 3D Bioimaging, European Molecular Biology Laboratory, European Bioinformatics Institute, Wellcome Genome Campus, Hinxton, United Kingdom

**Martin L Jones** is in the Electron Microscopy Science Technology Platform, Francis Crick Institute, London, United Kingdom

http://orcid.org/0000-0003-0994-5652

**Paul K Korir** is in Cellular Structure and 3D Bioimaging, European Molecular Biology Laboratory, European Bioinformatics Institute, Wellcome Genome Campus, Hinxton, United Kingdom

**Abraham J Koster** is in the Department of Molecular Cell Biology, Leiden University Medical Center, Leiden, The Netherlands

**Ingvar Lagerstedt** is in the European Molecular Biology Laboratory, European Bioinformatics Institute, Wellcome Genome Campus, Hinxton, United Kingdom

**Catherine L Lawson** is in the Center for Integrative Proteomics Research and the Research Collaboratory for Structural Bioinformatics, Rutgers, The State University of New Jersey, Piscataway, United States

**David Mastronarde** is in the Department of Molecular, Cellular, and Developmental Biology, University of Colorado, Boulder, United States

**Matthew McCormick** is at Kitware, Inc., Carrboro, United States

**Helen Parkinson** is in Molecular Archival Resources, European Molecular Biology Laboratory, European Bioinformatics Institute, Wellcome Genome Campus, Hinxton, United Kingdom

**Peter B Rosenthal** is in Structural Biology of Cells and Viruses, Francis Crick Institute, London, United Kingdom

**Stephan Saalfeld** is in at the Janelia Research Campus, Howard Hughes Medical Institute, Ashburn, United States

http://orcid.org/0000-0002-4106-1761

**Helen R Saibil** is in the Institute of Structural and Molecular Biology, Department of Crystallography, Birkbeck College, London, United Kingdom

**Sirarat Sarntivijai** is in Molecular Archival Resources, European Molecular Biology Laboratory, European Bioinformatics Institute, Wellcome Genome Campus, Hinxton, United Kingdom

**Irene Solanes Valero** is in the European Molecular Biology Laboratory, European Bioinformatics Institute, Wellcome Genome Campus, Hinxton, United Kingdom

**Sriram Subramaniam** is in the Laboratory for Cell Biology, Center for Cancer Research, National Cancer Institute, Bethesda, United States

**Jason R Swedlow** is in the Centre for Gene Regulation and Expression and the Division of Computational Biology, University of Dundee, Dundee, United Kingdom

**Ilinca Tudose** is in Molecular Archival Resources, European Molecular Biology Laboratory, European Bioinformatics Institute, Wellcome Genome Campus, Hinxton, United Kingdom

**Martyn Winn** is in the Scientific Computing Department, Science and Technology Facilities Council, Research Complex at Harwell, Didcot, United Kingdom

**Gerard J Kleywegt** is in the Molecular and Cellular Structure Cluster, European Molecular Biology Laboratory, European Bioinformatics Institute, Wellcome Genome Campus, Hinxton, United Kingdom

*Author contributions:* AP, Conceptualization, Resources, Supervision, Funding acquisition, Investigation, Visualization, Methodology, Writing—original draft, Project administration, Writing—review and editing, Organized workshop and chaired sessions; RB, DG, CH, MLJ, IL, CLL, DM, MM, SSaa, SSar, JRS, IT, Conceptualization, Writing—review and editing; SJB,

Conceptualization, Writing—review and editing, Chaired workshop sessions; LC, KG, JTH, AJK, PBR, Conceptualization, Writing—review and editing, Chaired workshop session; AI, HRS, SS, Conceptualization, Visualization, Writing—review and editing; PKK, Conceptualization, Software, Visualization, Methodology, Writing—review and editing; HP, Conceptualization; ISV, Conceptualization, Data curation, Writing—review and editing; MW, GJK, Conceptualization, Funding acquisition, Writing—review and editing

***Competing interests:*** SS: Reviewing editor, *eLife*. The other authors declare that no competing interests exist.

## Funding

| Funder | Grant reference number | Author |
|---|---|---|
| Medical Research Council | MR/L007835 | Gerard J Kleywegt |
| European Commission | 284209 | Gerard J Kleywegt |
| Wellcome Trust | 104948 | Gerard J Kleywegt |
| National Institutes of Health | R01 GM079429 | Gerard J Kleywegt |
| Medical Research Council | MR/N009614/1 | Martyn Winn |

The funders had no role in study design, data collection and interpretation, or the decision to submit the work for publication.

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
