## [Decision Letter]

Thank you for submitting your article "3D segmentation and transformation – building bridges between cellular and molecular structural biology" to *eLife* for consideration as a Feature Article. Your article has been reviewed by two peer reviewers, and the evaluation has been overseen by a Reviewing Editor and John Kuriyan as the Senior Editor.

The reviewers have discussed the reviews with one another and the Reviewing Editor has drafted this decision to help you prepare a revised submission

Summary:

The manuscript by Patwardhan et al., reports on the outcome of a workshop at the European Bioinformatics Institute (EBI) in Hinxton nr. Cambridge, UK. The workshop brought together experienced users and stakeholders from twelve different research laboratories and companies around the world. The purpose of the workshop was to come up with a roadmap for the annotation of structural data on the sub-cellular to cellular scale (primarily electron cryo-tomography and correlated light-EM (CLEM)), and to develop formats and tools for annotation that would be made available to the communities of structural and cellular biologists. Of particular interest will be tools for the integration of structural data from tomography, high-resolution cryoEM, protein crystallography and NMR and the development for a structured biological annotation tool for interactive annotation of tomographic and similar volumes, before uploading them to the EMDB data bank. This is a very worthwhile endeavour that will greatly benefit the future of structural and molecular cell biology. The manuscript itself is a valuable resource in that it provides numerous links to tools and websites that will be useful to many in the community.

Essential revisions:

1) Segmentation is considered from the perspective of structural biology, PDB and EMDB data, as well as sub-tomogram averaging. While integration of structures in the biological context is certainly a worthwhile effort, these are not true segmentation tasks. In fact, true segmentation tasks as in brain connectomics or cell biology are not mentioned at all.

2) The cross-correlation coefficient between a sub-tomogram average and the original volume is not useful as a means of quantification, as is well documented in the literature.

3) The article should distinguish more clearly between segmentation and template matching, and between annotations and metadata.

4) The expression "structured biological annotation" is used repeatedly in the manuscript but is difficult to understand. The authors should explain what they mean more clearly.

---

## [Author Response]

*Essential revisions:*

*1) Segmentation is considered from the perspective of structural biology, PDB and EMDB data, as well as sub-tomogram averaging. While integration of structures in the biological context is certainly a worthwhile effort, these are not true segmentation tasks. In fact, true segmentation tasks as in brain connectomics or cell biology are not mentioned at all.*

We agree that the scope of the term segmentation and what it entails for different imaging fields can vary. We have now added a sentence explicitly explaining this to be the case and that the manuscript focuses on the narrower definition outlined in the manuscript.

*2) The cross-correlation coefficient between a sub-tomogram average and the original volume is not useful as a means of quantification, as is well documented in the literature.*

We agree that the scope of the term segmentation and what it entails for different imaging fields can vary. We have now added a sentence explicitly explaining this to be the case and that the manuscript focuses on the narrower definition outlined in the manuscript.

*3) The article should distinguish more clearly between segmentation and template matching, and between annotations and metadata.*

The manuscript as such does not deal with template matching but with relating sub-tomogram averages to tomograms. The file format could be used to deal with template matching and a line has been added explaining this. We have also gone through the manuscript and updated a couple of instances where it would be more appropriate to use the term ‘meta-data’ rather than annotations.

*4) The expression "structured biological annotation" is used repeatedly in the manuscript but is difficult to understand. The authors should explain what they mean more clearly.*

In the original manuscript we had defined “Structured biological annotation” explicitly as the first sentence in the section on “Structured biological annotation”. The term was used earlier on in the text and we understand that this could have been the source of confusion. We now define “Structured biological annotation” early on in the text before it is first used.